# A modular fluorescent camera unit for wound imaging
Maryam Tebyani [1,2] ✉, Gordon Keller[1], Wan Shen Hee [1], Prabhat Baniya [1], Alex Spaeth [1,2], Tiffany Nguyen[1], Harika Dechiraju[1], Anthony Gallegos[3], Héctor Carrión [4], Derek Hamersly[5], Cristian Hernandez[1], Alexie Barbee [1], Hao-Chieh Hsieh[1], Elham Aslankoohi[1], Hsin-ya Yang [3], Narges Norouzi[6], Min Zhao [3,7], Alexander Sher [5], R. Rivkah Isseroff [3], Marco Rolandi [1,2] ✉ & Mircea Teodorescu [1,2] ✉

Advanced imaging tools are revolutionizing the diagnosis, treatment, and monitoring of medical conditions, offering unprecedented insights into live cell behavior and biophysical markers. We introduce a modular, hand-held fluorescent microscope featuring rapid set-up and sub-millimeter resolution for real-time biological analysis. We apply our system to map pH and nitric oxide (NO), biomarkers central to wound healing, in subcutaneous wounds. Using machine learning to cluster pH reveals spatiotemporal trends, including a concentric gradient peaking at the center and stabilization at the wound edge. NO clustering shows high-concentration structures that decrease in size but intensify as healing progresses from hemostasis to proliferation, enabling prediction of the healing day and re-epithelialization. These biomarker mappings offer insights poised to inform future wound healing studies. This research lays the groundwork for integrating the modular imaging unit with bioelectronic devices in closed-loop feedback systems, using machine learning to guide optimal wound treatment and accelerate healing.

Imaging tools play an integral role in biological, biomedical, and clinical settings, offering valuable insights for diagnosis, treatment planning, and monitoring[1]. Light microscopy is a powerful, non-invasive, quantitative measurement tool in biology, allowing for the analysis of molecules and cells[2]. Fluorescence microscopy further amplifies our understanding of biological systems by providing deeper insights through the use of biomolecular markers. Applications of fluorescence imaging extend to live cell tracking, cancer detection[3], virus detection[4], evaluating food safety, and guiding robotic-assisted surgery[5,6].

While traditional fluorescent microscopes are often bulky and expensive, recent advances in image sensors, lenses, and emission filters have paved the way for portable units that can be used in more versatile environments[7–10]. Many of these systems employ integrated operating systems, leveraging commercial off-the-shelf digital cameras or smartphones. This approach results in portable, affordable, and adaptable fluorescent microscopes. However, these operating systems can impose limitations on available functionalities in comparison to a dedicated microcontroller.

Microcontrollers offer the advantage of flexible onboard computing along with more comprehensive power and control options for complementary subsystems. In previous work, portable imaging systems based on Arduino or Raspberry Pi platforms have been developed, incorporating both brightfield and fluorescent microscopy functionalities[11–14]. These systems are often low-cost and highly adaptable, benefiting from rapid manufacturing processes like 3D printing and the use of low-cost, commercial-off-the-shelf components.

Monitoring in vivo fluorescence can provide insights into physical or chemical changes at the cellular or tissue level, achieved by detecting biomarkers that vary in response to pathological alterations[15]. Hand-held, portable fluorescent microscopy systems are well-suited to in vivo imaging due to irregular surfaces and imaging constraints, like the physical size of samples. One quickly changing in vivo environment is that of a wound bed, which requires timely monitoring to support clinical decision-making that promotes healing[16]. Wound healing is a dynamic and complex process comprised of overlapping stages: hemostasis, inflammation, proliferation,

[1]Department of Electrical and Computer Engineering, University of California Santa Cruz, Santa Cruz, CA, USA. [2]Genomics Institute, University of California Santa Cruz, Santa Cruz, CA, USA. [3]Department of Dermatology, School of Medicine, University of California Davis, Sacramento, CA, USA. [4]Department of Computer Science and Engineering, University of California Santa Cruz, Santa Cruz, CA, USA. [5]Santa Cruz Institute for Particle Physics, University of California Santa Cruz, Santa Cruz, CA, USA. [6]Department of Electrical Engineering and Computer Science, University of California Berkeley, Berkeley, CA, USA. [7]Department of Ophthalmology & Vision Science, University of California Davis, Sacramento, CA, USA. ✉e-mail: mtebyani@ucsc.edu; mrolandi@ucsc.edu; mteodore@ucsc.edu

and maturation[17]. This imaging modality provides a platform for tracking the progress of wound healing by monitoring pivotal features like biomarkers[15,18–21], innervation[22], and epithelialization[23]. Previously, brightfield cell phone images have been used to classify wound stage[24,25]. Further, researchers have developed electronic bandages that can deliver specific drugs to wound beds[26]. These smart bandages have been proposed for closed-loop wound healing, where measurements from integrated sensors are used to direct drug delivery[27]. Combining brightfield and fluorescent imaging for wound stage monitoring to control drug delivery offers a method for a closed-loop feedback control system based on informed models for expedited wound healing and closure[28].

In this paper, we present a hand-held and modular imaging unit, shown in Fig. 1, that can be used for biological imaging. Our imaging unit consists of five fundamental components, which can be customized to accomplish an array of imaging tasks, including brightfield imaging and interpreting various fluorescent dyes. Key metrics of our design include hand-held usage, ease of operation, short set-up time of a few minutes, 6 mm field of view, sub-millimeter resolution, and RGB/fluorescent imaging modalities. We begin by giving a brief overview of the imaging unit and system architecture. Next, we present the results collected with the developed system. We show brightfield microscopy usage through images of a calibration slide and fluorescent microscopy through images of stained

THP-1 cells and macrophages. We demonstrate the use of the imaging unit for pH and nitric oxide (NO) detection, both in vitro and in vivo, along with an analysis of the detection using machine learning. Finally, we present the design and development of our system, the imaging procedure in vitro and in vivo, and experimental details. We show that our system can successfully resolve sub-millimeter biological features and detect relative changes in pH and NO in vivo, which are key biomarkers in wound healing.

## Results and discussion

The imaging unit, a modular fluorescent camera system, has been designed for flexible assembly and operation. Figure 1a shows the primary use case for the imaging unit, where a wound image is captured and analyzed to estimate features like the wound stage, re-epithelialization, and pH. In this section, we present the system design and images resulting from the development of the imaging unit. Further, we present an analysis and discussion of the imaging results.

### System design

Our system is constructed using commercial-off-the-shelf components, including a compound lens, filter, camera, Raspberry Pi, a light emitting diode (LED) printed circuit board (PCB), and a silicone spacer that doubles as a light diffuser, as shown in Fig. 1b. These components are integrated

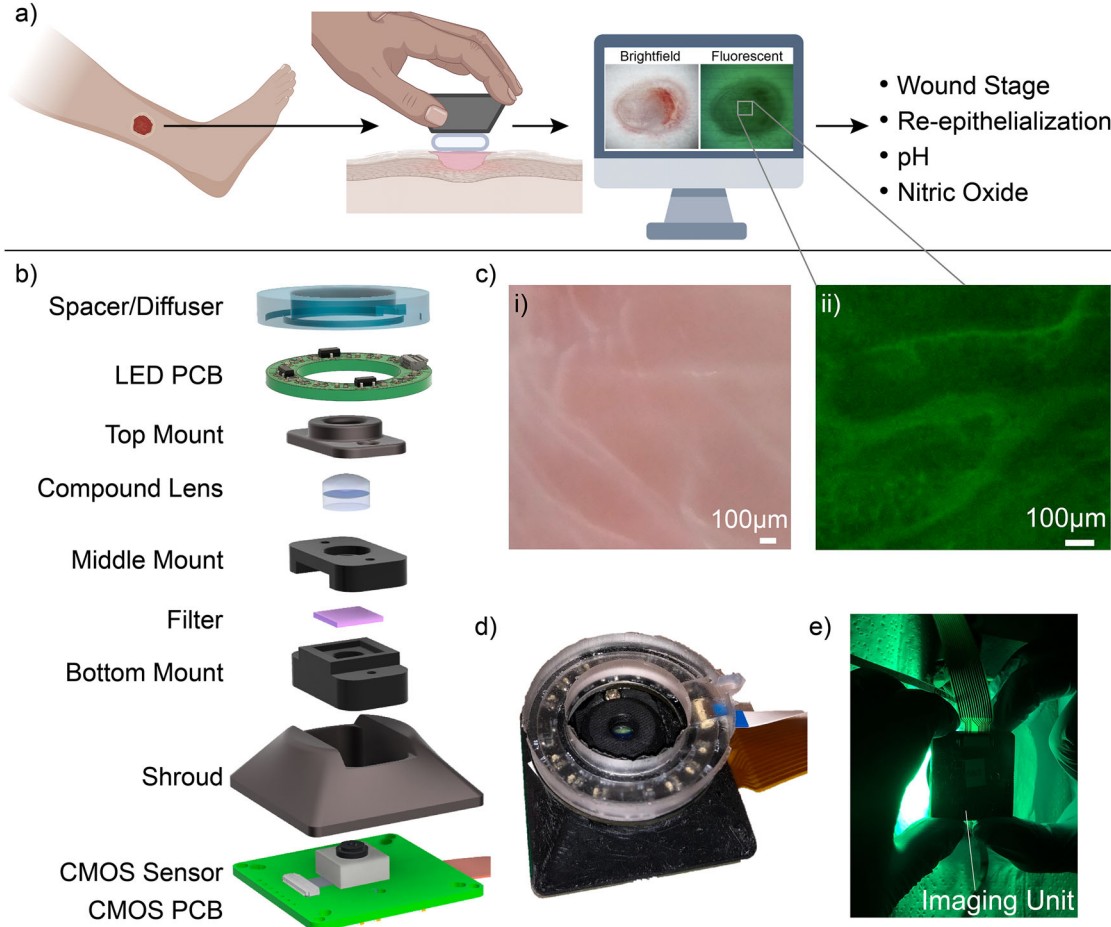

**Fig. 1 | Overview of the imaging unit.** Demonstrates proposed usage and composition of the imaging unit, along with example images. **a** A primary use case for the imaging unit. Brightfield and fluorescent wound images can monitor the wound stage, re-epithelialization rate, pH, and nitric oxide levels[57]. **b** The components required for the fluorescent configuration of the device. The camera, filter, lens, 3D printed components, LED PCB, and spacer/diffuser are shown in an isometric, expanded view. For brightfield imaging, the filter is excluded. LED PCB ribbon cable not pictured. **c** In vivo images captured by the imaging unit, zoomed in to highlight

features within the wound bed, including the fluorescent image displayed on the computer monitor in (**a**). (i) Brightfield image of an in vivo excisional skin wound on the dorsum of a mouse, with a 100 μm scale bar. (ii) Fluorescent image of the wound stained by DAF-FM with a 100 μm scale bar. **d** The assembled imaging unit, including the camera shroud, LED PCB, and transparent diffuser, along with two ribbon cables required for communicating with the camera and LED PCB. **e** The bottom side of the device is shown as a user performs hand-held fluorescence imaging in vivo.

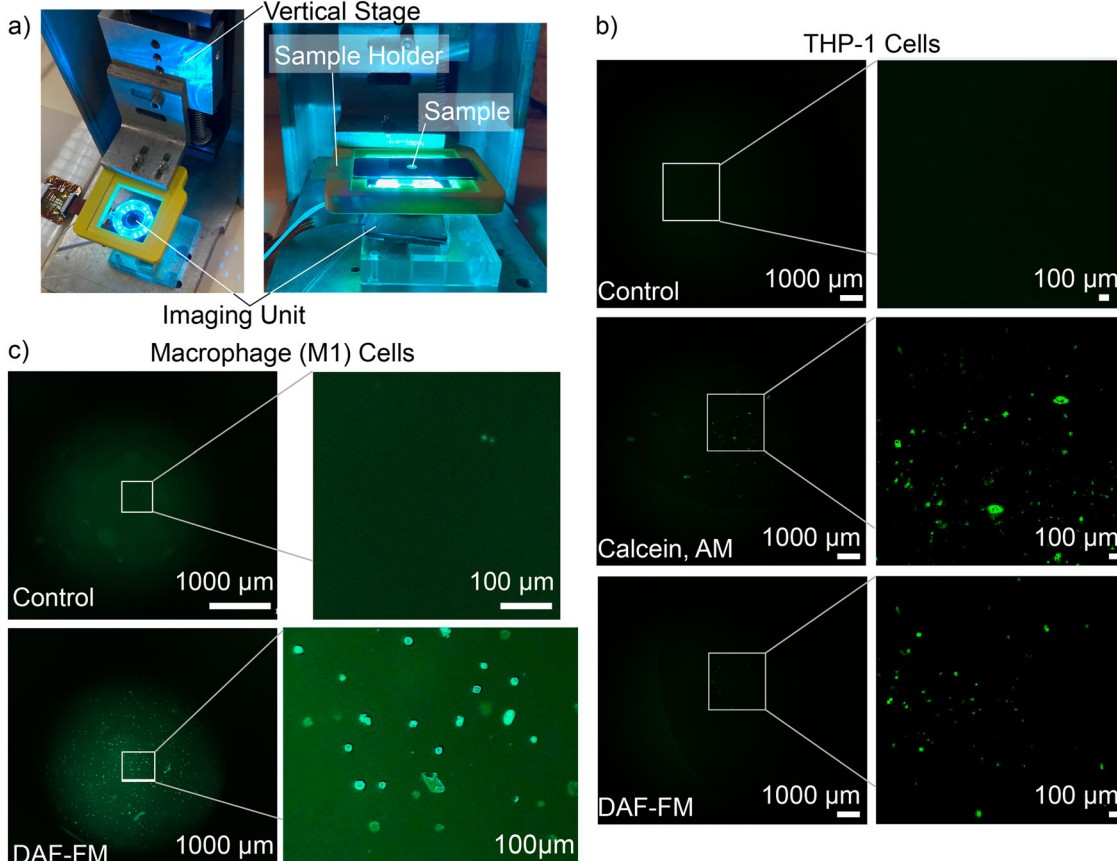

**Fig. 2 | Calibration setup and fluorescent cell images captured with our imaging unit. a** Calibration and in vitro imaging setup. The imaging unit is placed face up, and a linear stage controlled by a dial is used to hold and position the sample. **b** Fluorescent imaging of THP-1 cells. The full image is shown with a 1000 μm scale bar, and the inlet is zoomed in with a 100 μm scale bar. The inlet images are color-mapped to increase the contrast of the fluorescent signal. Control: control slide with no dye. Calcein, AM: cells dyed with Calcein AM fluorescent compound. DAF-FM: cells dyed with DAF-FM fluorescent compound. **c** Fluorescent imaging of macrophage (M1) cells. The full image is shown with a 1000 μm scale bar, and the inlet image is zoomed in with a 100 μm scale bar. Control: control slide with no dye, DAF-FM: cells dyed with DAF-FM fluorescent compound.

using a modular 3D-printed housing (designed with Autodesk Fusion 360[29]) which can be adjusted to accommodate various lenses, filters, and camera systems. Figure 1c shows magnified sample in vivo wound images, Fig. 1d shows an assembled imaging unit, and Fig. 1e shows its handheld usage. In the presented design, we use an Aspherized Achromatic Compound Lens to reduce spherical and chromatic aberrations, enabling imaging of relevant biological systems[30]. The ring-shaped LED PCB hosts 12 LEDs for sample illumination and fluorescent excitation, and the silicone spacer, or diffuser, is designed to be positioned on top of the LED PCB. The silicone diffuser stabilizes the camera during imaging and is made of polydimethylsiloxane silicone (PDMS), a biocompatible material. Optional filters can be mounted within the printed housing to isolate specific emission wavelengths of fluorescent signals. We use a Raspberry Pi and Arducam 21MP CMOS sensor as our computing and camera system. This combination provides a flexible user interface, enables programmable camera usage, and allows real-time image analysis. We use libcamera, a software library that supports camera systems directly from the Linux operating system[31]. The Arducam 21 MP camera has an integrated motorized camera lens that can capture a variable focus parameterized on a scale from 0 to 1000. This scale corresponds to the object distance, which varies depending on the specific imaging setup.

**Imaging**

In this report, we use the imaging unit for brightfield imaging and, with various LED and filter configurations, image three fluorescent dyes (DAF-FM, Calcein AM, Snarf). Figure 1c presents sample images from our unit,

showing a zoomed-in section within the wound bed, including the fluorescent image from the boxed region shown in Fig. 1a. The left image shows a brightfield image obtained in vivo from an excisional wound on the dorsum of a mouse, while the right panel shows a fluorescent image of the same in vivo wound stained with DAF-FM. The imaging unit can be used handheld, as shown in Fig. 1e, or hands-free, such as in conjunction with an optical stage, shown in Fig. 2a. To ensure images are in focus during in vitro and in vivo imaging, we first determine the focal distance of the imaging unit through a calibration process. This is done by using the optical stage setup (Fig. 2a) with an adjustable object distance. A calibration slide (Edmund Optics USAF 1951 Glass Slide Resolution Target), shown in Supplementary Fig. S1, is placed on the linear stage, with the imaging unit positioned at the bottom of the stage. The micrometer screw dial is adjusted until the calibration slide is in focus. The measured distance between the microscope slide and the imaging unit is then recorded as the focal distance. The spacer/diffuser element is designed with a height calculated as the difference between the height of the imaging unit and this experimentally determined focal length. The spacer/diffuser height ensures the imaging unit is positioned at the correct distance from biological samples during imaging.

**Cell imaging.** The calibration setup, shown in Fig. 2a, is also used to image microscope slides with fluorescent-dyed cells. Figure 2b shows THP-1 cell imaging using the lens configuration shown in Fig. 1b, with a zoomed-in version where the fluorescent signal has been color-mapped to improve visualization. The Control image has no dye applied; the Calcein, AM image shows the THP-1 cells dyed with Calcein, AM

**Fig. 3 | Imaging unit calibration for pH level detection using a fluorescent dye with a wound proxy. a** Calibration setup diagram using the imaging unit face down and positioned atop a polydimethylsiloxane (PDMS) wound proxy. The fluorescent sample is pipetted into a well that simulates the wound bed within the PDMS proxy. The ribbon cables are not pictured. **b** Calibration results for Snarf, a pH-sensitive fluorescent dye, buffered from pH 5 to 9. The pH is shown on the *x*-axis, and the fluorescent intensity, represented by the red color channel, is shown on the y-axis.

The line represents the calibrated linear model for pH detection using the mean fluorescent intensity, indicated by red dots. The distribution of the fluorescent intensity in each cropped image ($n$ = 858,407 pixels) is shown as box plots, where the center line denotes the median, the box spans the interquartile range (IQR), and the whiskers extend to the most extreme data points within 1.5 × IQR. **c** Cropped images from the imaging unit at each pH level used for the calibration.

fluorescent compound, and the DAF-FM image is dyed with DAF-FM fluorescent compound. Calcein AM, a cell-permeant fluorescent dye, reports cell viability through whole-cell imaging. DAF-FM, also a cell-permeant fluorescent dye, reports the presence and quantity of intracellular NO. The THP-1 cell line has been used extensively to study monocyte/macrophage function and NO signaling[32], which is relevant to the inflammatory response of the in vivo NO experiment presented in this report. Figure 2c shows bone marrow macrophage (M1 type) cell imaging using a different lens configuration, shown in Supplementary Fig. S2. The Control image has no dye applied, and the DAF-FM image shows the cells stained with DAF-FM fluorescent compound. The filter and LED configuration, along with additional details for the fluorescent dyes, are described in the Methods section.

**pH detection**. In this section, we show the use of the imaging unit for pH detection of an in vivo wound bed through fluorescent imaging. Wound pH is integral to the healing process and influences various biochemical reactions essential for tissue repair. The physiological pH of healthy skin is slightly acidic, typically ranging from 4.5 to 5.3, which has been shown to promote optimal healing conditions[19]. However, wounds tend to shift to a more alkaline environment, with chronic wounds exhibiting pH levels ranging from 7.42 to 8.9. This alkaline state is conducive to bacterial growth, which can impede healing and increase the risk of infection[19]. Conversely, maintaining an acidic pH, around pH 4, has been shown to improve healing rates and epithelialization, along with promoting collagen synthesis[19]. Studies have also highlighted the importance of specific pH ranges during distinct phases of acute wound healing, where the pH initially decreases and then increases during the inflammatory phase, continues increasing throughout granulation or the proliferative phase, and finally, gradually returns to the acidic state of healthy skin[18]. Thus, monitoring wound pH is likely a valuable therapeutic strategy in wound management to facilitate faster recovery and reduce complications associated with infection[33]. Previously, a disposable device was developed to map wound pH by pressing it directly on the wound surface[34].

In this work, we demonstrate the capability of the imaging unit to remotely and non-invasively map wound pH. First, we calibrate the imaging unit's pH detection and compare the results to a commercial fluorescent microscope. Next, we employ this calibration to map the pH of an in vivo subcutaneous wound bed. Finally, a machine-learning algorithm, *k*-means clustering, is applied to segment the wound for visualization of pH dynamics within individual wound beds.

We use pH buffers to validate the imaging unit's fluorescence detection capabilities and compare the response to a commercial fluorescent microscope. To build an accurate model of the in vivo fluorescence response, we

created a proxy of the expected wound geometry with PDMS, with dimensions shown in Supplementary Fig. S3. Carboxy SNARF-1 (Snarf), which is excited at wavelengths between 488 nm and 530 nm and emits fluorescence at 580 nm and 640 nm, was used as the pH-sensitive dye. Figure 3a shows the experimental setup used to capture images of the Snarf dye, diluted across a range of biologically relevant pH buffers for early-stage acute wound healing (pH 5–9)[19]. Although pH values below this range may have therapeutic value or be relevant to late-stage wound healing in vivo, they fall outside the linear range of the fluorescent response, as shown in Supplementary Fig. S4a. Figure 3b shows the fluorescent intensity for the linear range, which is measured by the red color channel (640 nm wavelength) of each image shown in Fig. 3c, as a function of pH level. The boxplots demonstrate the variation in the fluorescent intensity across the images (858,407 pixels). Figure 3c shows the cropped section of the resulting images captured by our unit used to build the calibration. To characterize the effect of pH on fluorescent intensity, we used a linear regression model[35] to fit the data, yielding the relationship $FI = 28.8 \times pH - 120.05$ with $R^2 = 0.957$. Table 1 details the percent error between the estimated and actual pH, calculated using this linear regression model for the imaging unit and the ratiometric method for the commercial microscope as detailed in the Supplementary Information (SI). The percent error is calculated by taking the absolute difference between the actual and estimated pH values, dividing it by the actual pH, and then multiplying by 100 to express the error as a percentage. The accuracy of the imaging unit is comparable to that of the commercial microscope for this imaging task, as further shown in Supplementary Fig. S4b. However, we note that the imaging unit's error rate is larger at the higher pH values. Although the overall emission intensity at the 640 nm wavelength increases with pH, the reported fluorescent emission for this dye increases less with pH at higher pH values. As shown in the fluorescent intensity plot in Fig. 3b, the change in signal between pH 7 and 8 is greater than between pH 8 and 9, resulting in reduced sensitivity in the higher pH range. While our linear regression model achieves comparable accuracy to the ratiometric method, future studies could collect additional calibration data at intermediate pH values to improve this method.

**Table 1 | Percent error of estimated vs. measured pH for the commercial microscope and the imaging unit Snarf dye calibration**

| pH | 5.2 | 6.0 | 7.1 | 7.9 | 9.0 |
|---|---|---|---|---|---|
| Commercial | - | 4.4% | 5.4% | 4.1% | - |
| Imaging unit | 3.1% | 1.5% | 1.4% | 6.2% | 3.5% |

The extreme pH is not available for the commercial microscope due to the ratiometric calibration method.

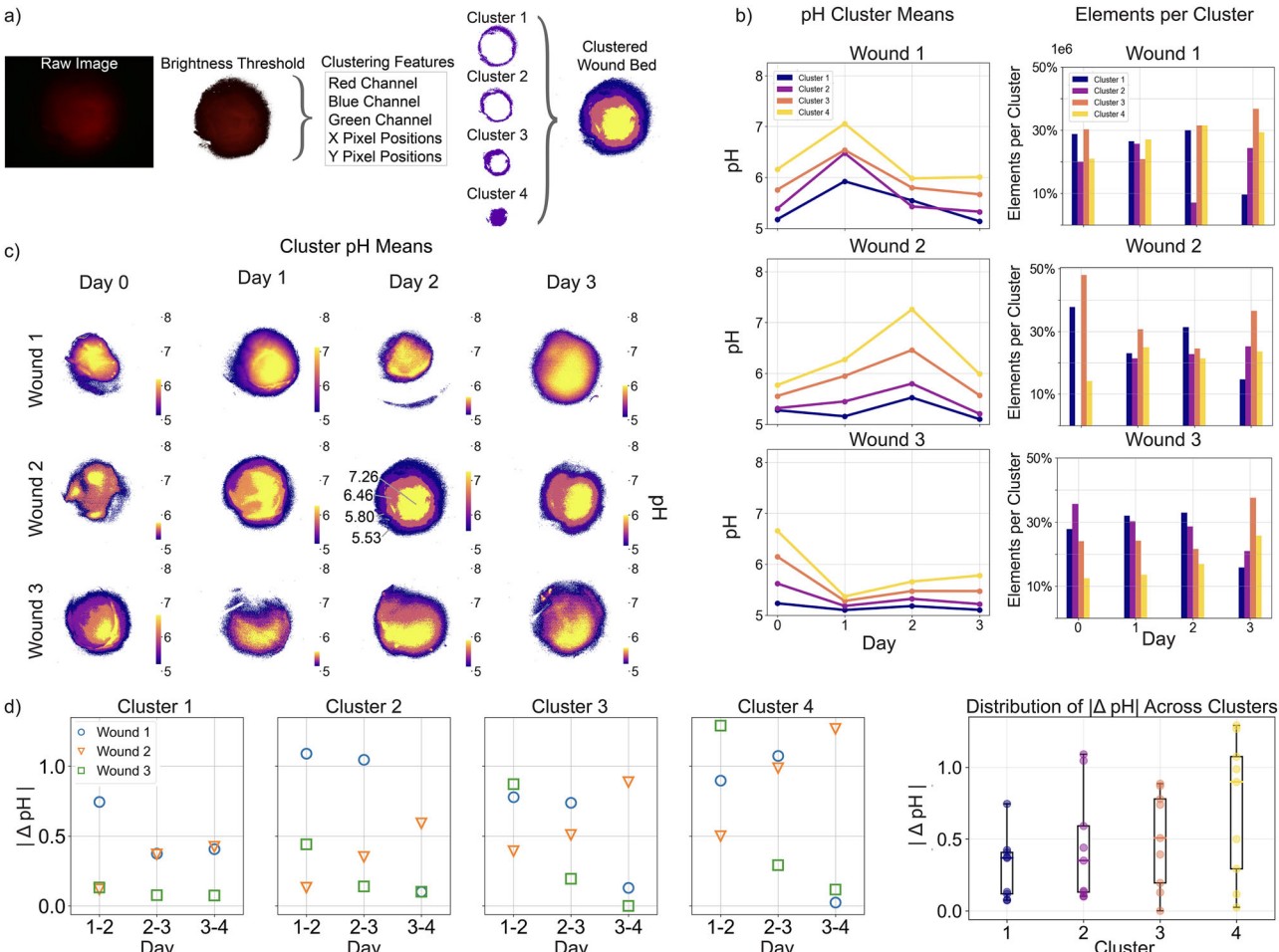

**Fig. 4 | pH analysis using *k*-means clustering for in vivo wound bed image segmentation.** The wound bed pH clusters are tracked over a 4-day experiment. **a** This figure illustrates the sequence for visualizing the segmented wound. The first step is to load the raw fluorescent image, where the pixel intensity in the red color channel is directly proportional to pH. To this image, a brightness threshold is applied to eliminate the dark background. Next, the clustering features for classification are assembled, including the red, green, and blue color channels, along with the *x* and *y* pixel positions. These features are input to the *k*-means clustering algorithm with *k* = 5. Cluster 0 contains the background pixels and is excluded from the remainder of the figure. Finally, each cluster is colored based on its mean pH. **b** These plots demonstrate the dynamics of the wound clusters. The pH cluster means and number of elements per cluster, or how many pixels are in each cluster, are shown as a function of the experimental day. **c** This figure displays the clustered wound beds based on the sequence depicted in (**a**). The three wounds are shown over the 4-day experiment, segmented into four clusters. The color bars indicate pH and are shown individually to highlight variations within the wound bed. **d** These plots show the magnitude of pH change throughout the experiment for each cluster. The subplots visualize the magnitude of pH change between consecutive days for each cluster. The information is also represented as a boxplot, where the *x*-axis is the cluster index and the *y*-axis is the magnitude of pH change. The variation and mean in the magnitude of pH change are higher in cluster four, which is generally at the wound center.

A two-filter system could also be developed to support the ratiometric method. Details regarding the pH buffers and fluorescent imaging are provided in the "Methods" section.

A 4-day in vivo mouse model experiment was performed with our imaging unit to detect the pH of a 6 mm diameter wound using Snarf fluorescent dye. Images were captured daily in *z*-stacks of 11 images, with stage height evenly spaced from 0 to 1000. To gain insight into the pH dynamics of the wound, we analyzed the resulting image set using *k*-means clustering, an algorithm designed to partition *n* observations into *k* clusters[36]. Previously, this algorithm was used to remove the background of brightfield wound images and isolate the wound[37]. Our approach instead focuses on clustering the wound itself into various regions. For this study, we use *n* = 11 observations corresponding to the *z*-stack images and set *k* = 5. The selection of *k* is numerically determined, with further discussion available in the SI and Supplementary Fig. S5. The image processing, depicted in Fig. 4a, begins by applying a brightness threshold to the raw image, followed by extracting the clustering features, which include the RGB color channels and the *x*, *y* positions for all pixels. Here, the raw fluorescent image shown appears more diffuse than the RGB and NO imaging results. This is likely due to the inherent nature of pH imaging; as H+ ions diffuse to adjacent tissues, we do not expect to see specific biological structures in the wound. While only the red color channel is used for the final pH mapping, the blue and green color channels have a small signal due to filtering and contribute some information about the wound topology. The *k*-means algorithm[38] then processes these features to classify each pixel into one of *k* groups, denoted by Cluster 1 through Cluster 4. Cluster 0 represents the pixels below the brightness threshold and is not explicitly of interest for the remaining analysis. We color each cluster based on the cluster's mean pH, as depicted in the Clustered Wound Bed in Fig. 4a. Figure 4b shows the clustering trends across the three wounds over the 4-day experiment and Fig. 4c shows the clustering outcome. The color bar in Fig. 4c shows the range of pH values for each wound bed, highlighting variations in pH for each sample. Wound 2 on day 2 features call-outs for the mean pH values of each cluster.

The pH values captured with our imaging unit, summarized in the pH Cluster Means column of Fig. 4b, align with previously reported ranges.

Literature has shown that acute wounds range from high pH 5 to mid-pH 6 during the inflammatory phase[18], and have a mean pH value of 7.44 throughout healing[19]. Fig. 4b shows variability in both the direction of pH value changes and elements per cluster between days and samples. While absolute pH values are understood to vary between healing (and non-healing) wounds[39], differences in deflection direction suggest variations such as the timing of inflammatory processes. The bar plots showing elements per cluster demonstrate corresponding differences in wound structure and composition, as represented by cluster distributions. Clustering variability may be due to differences in the wound shape between samples and days. The pH mapping in Fig. 4c shows the entire wound. The wounds variably exhibit increases in wound area as the days progress, such as wound 1 from day 0 to 1, which has also previously been reported in similar wound models[24].

The resulting wound clustering produces concentric circles, with the highest pH values at the center, decreasing outward. This pattern is also evident in the pH Cluster Means column of Fig. 4b, which shows that Cluster 4 (yellow) consistently exhibits the highest mean pH across all samples.

This analysis method facilitates tracking pH trends, such as in Fig. 4d, where we observe that the rate of change in pH is more stable at the edges of the wound (Cluster 1) and more varied at the center of the wound (Cluster 4). In particular, we plot each cluster's magnitude change in pH ($|\Delta pH|$) across all wounds to study the mean and standard deviation as a function of day and cluster, where the clusters are generally distributed as concentric rings. The standard deviation for $|\Delta pH|$ is observed to be highest for the center cluster and lowest for the wound edge cluster: 0.21, 0.37, 0.32, 0.46 for Cluster 1–Cluster 4, respectively. Statistical analysis confirms this, as the variance of $|\Delta pH|$ at the wound center is significantly greater than in the peripheral regions (Clusters 1–3) with a chi-squared test for variance resulting in a one-sided $p$-value of 0.031 with $\chi^2(8) = 16.95$. The variance of $|\Delta pH|$ is 0.04, 0.14, 0.10, 0.21 for Cluster 1–Cluster 4, respectively. This observation is consistent with a previously derived computational model, which demonstrated that wound pH is significantly influenced by the wound size[40]. The computational model indicated that larger wounds have higher rates of change in pH, primarily due to a boundary term in their formulation. Essentially, the pH at the wound edges is expected to change more slowly due to the stabilizing influence of the surrounding tissue, whereas the pH at the wound center is more variable. In our experimental results, we observe that the center cluster indeed consistently exhibits the highest standard deviation or spread in the magnitude of pH change, along with a higher average rate of change across the dataset. This supports the theoretical conclusion of the computational model that wound edges maintain a more constant pH compared to the wound center. This variability in the pH of the wound center may influence the healing process and the effectiveness of wound treatments.

In summary, the imaging unit's ability to detect biologically relevant pH values was validated by characterizing its response and comparing it to a commercial fluorescent microscope using a wound proxy. We derived a linear model of fluorescence response as a function of pH, which was then used to estimate the pH in three experimental wounds across 4 days. The *k*-means clustering algorithm enabled segmentation of the wound, revealing a general trend of the wound dynamics: higher pH levels and variation towards the wound center. In the future, comparing the imaged wound pH to an average value obtained using traditional measurement methods could offer insights into the relationship between these approaches. Continued imaging until full wound closure could also provide valuable insights into pH dynamics throughout the entire healing trajectory. A two-filter system could also be introduced, which maintains the portability and compact form factor of the imaging unit through a 3D-printed mechanism integrated into the housing, switching the position of the enclosed filters. This could further enable simultaneous imaging of biomarkers, as discussed in the following section. Ultimately, these images could be input to machine learning models that integrate clinical, biochemical and molecular characterization of the wound to generate an algorithm that could determine wound state and predict healing trajectory.

**Nitric oxide (NO) and wound stage detection.** In this section, we show the use of the imaging unit for NO detection of an in vivo wound bed through fluorescent imaging. NO assists in regulating multiple aspects of wound healing. It mediates vascular homeostasis[41], inflammation[42], and antimicrobial action[42,43]. During the inflammatory phase, NO is produced by inducible nitric oxide synthase to coordinate the immune response and clear pathogens[21,42,44]. NO influences cell proliferation, collagen formation, and wound contraction[42], along with promoting angiogenesis, essential for tissue repair and regeneration[45]. NO also helps regulate cytokines that initiate inflammation and modulates the migration and attachment of neutrophils to the endothelium[41,42]. In diabetic wounds, which are often chronic, NO donors have shown potential to partially reverse the impaired healing process[42]. The various functions that the NO molecule mediates in wound repair render it critical for effective healing and a promising candidate for therapeutic interventions.

Here, we demonstrate the use of DAF-FM dye for imaging NO levels in an in vivo wound bed. DAF-FM has been shown to be a reliable indicator of NO in various biological contexts[46]. DAF-FM has a fluorescence quantum yield of about 0.005, which increases to about 0.81, ~160-fold, after reacting with NO. Although DAF-FM can enter and stain dead cells, the significant difference in the quantum yield indicates that the imaged fluorescent signals will primarily reflect NO concentration rather than nonspecific staining. Previously, we demonstrated the efficacy of the imaging unit for use with DAF-FM fluorescent dye, shown by cell imaging in Fig. 2b, c. In this experiment, we captured brightfield and fluorescent images of eight wound beds on four separate mice on sequential days post-wound generation. Specifically, two unsplinted 6 mm wounds were made on the back of each mouse. One mouse was wounded per day on days 0, 1, 2, and 3, and the experiment was concluded on day 6, providing histology data for days 3, 4, 5, and 6. Due to time constraints of in vivo imaging, non-stained control fluorescent images were not captured. After imaging, the animals were sacrificed, and the wounds were excised and fixed for histological analysis. Histological analysis of the wounds was conducted to determine the percent re-epithelialization (% re-epi.) for each wound. We use re-epithelialization as a metric for assessing wound healing progress because it measures the restoration of the epidermis, the skin's protective barrier, which indicates how effectively the wound is closing[23].

Two sample wounds are shown in Fig. 5a, with 1 mm scale bars. Figure 5ai shows representative brightfield imaging unit pictures, which were captured before the DAF-FM dye was applied. Figure 5aiii shows fluorescent images of the wounds captured after the dye was applied to the wound bed. Zoomed-in regions of the brightfield and fluorescent wound images are shown in Fig. 5aii, aiv, converted to grayscale for easier comparison. In the fluorescent image, areas with brighter pixels correspond to higher NO levels. Visual inspection reveals bright, linear structures in the fluorescent images, indicating increased fluorescence intensity. This is highlighted in the zoomed-in images in Fig. 5aiv. We hypothesize that these structures are capillaries because endothelial cells, which form the lining of capillaries, produce NO. The presence of these linear structures in fluorescent images but not in brightfield images suggests that they are not merely reflections of wound topography. Figure 5aiv also shows that regions of the wound have varying amounts of capillaries. The full set of wound images is available in Supplementary Fig. S6.

To demonstrate the utility of this imaging technique, the 8 brightfield images captured were used to predict the wound stage probability, and the corresponding fluorescent images to create a model for predicting the day post-wound generation and the % re-epi. Table 2 shows the results of applying Healnet[25] to predict the wound stage probability based on the brightfield images for each wound. Healnet is a previously developed machine learning algorithm developed to classify wound

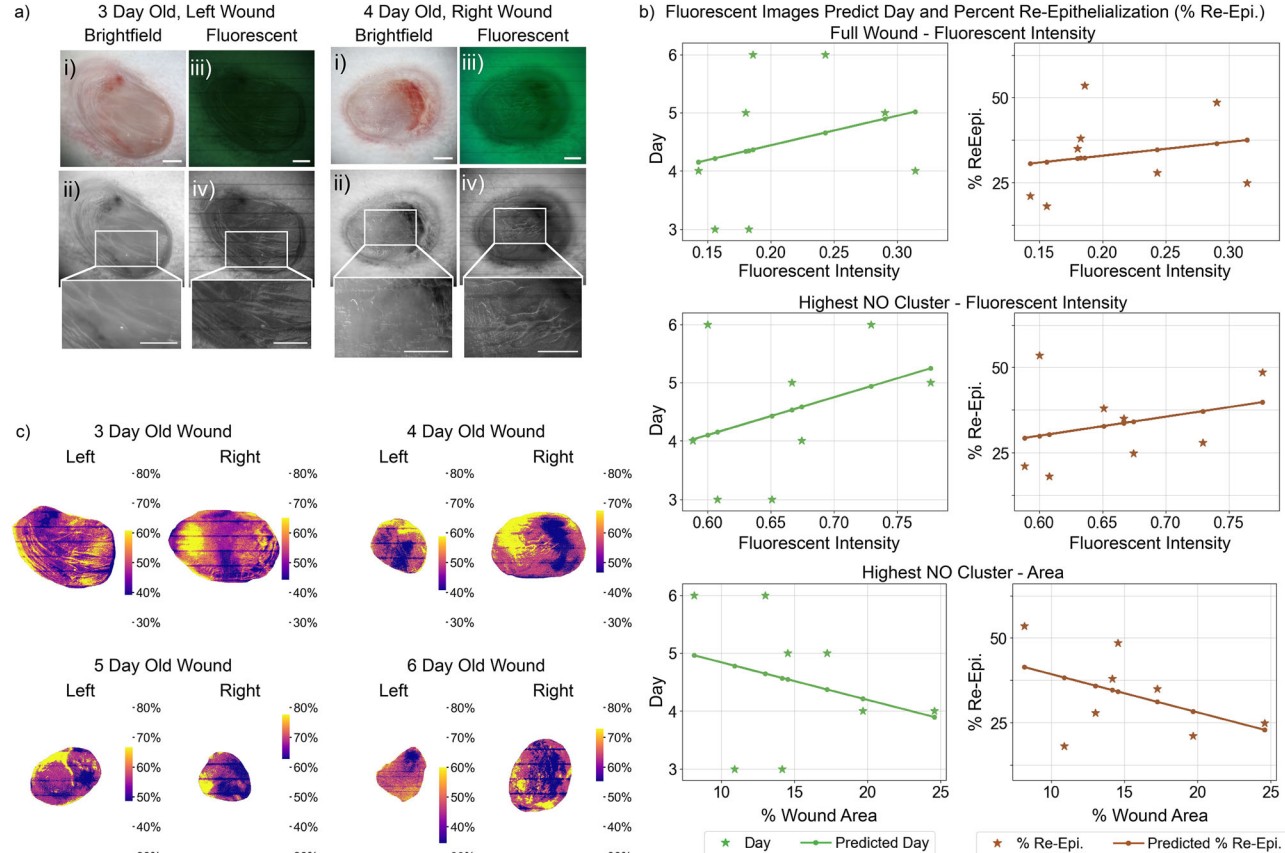

**Fig. 5 | Day post-wound generation and re-epithelialization prediction based on in vivo fluorescent images of nitric oxide (NO). a** Brightfield and fluorescent pictures dyed with DAF-FM captured by the imaging unit. Scale bars in the bottom right corner are 1 mm. (i) Brightfield image of the wound bed. (ii) The brightfield image shown in grayscale with a zoomed-in region. (iii) Fluorescent image of a wound bed dyed with DAF-FM. (iv) The fluorescent image is shown in grayscale with a zoomed-in region. **b** The fluorescent images are used to predict the day of healing and the percentage of re-epithelialization (% re-epi.) using a regression model. The full wound—fluorescent intensity plot shows the wound data and predictions based on the mean fluorescent intensity of the wound bed, which has a predicted positive relationship. The highest NO cluster-fluorescent intensity plot

shows results based on $k$-means clustering, using the cluster with the highest mean fluorescent intensity in the clustered wound bed, which also has a positive relationship. The highest NO cluster—area plot shows the results based on the percentage of the wound area of the same wound cluster, which has a negative relationship. Our findings imply that as wound healing progresses, the area of the wound that highly expresses NO decreases, even though the intensity of NO within these regions remains elevated. This suggests that the spatial distribution of NO expression, rather than its overall intensity, may play a critical role in wound healing. **c** These images show the $k$-means clusters for each experimental wound. The color bars indicate the NO fluorescent intensity as percentages and are shown individually to highlight variations within the wound bed.

stages — hemostasis, inflammation, proliferation, and maturation - by assigning probabilities to each stage given a brightfield wound image. The algorithm predicts that the right 3- and both 4-day-old wounds are in hemostasis with a high probability. The left 3-day-old wound probability is split between hemostasis and inflammation. The day 5 wounds and the left day 6 wound are predicted to be in proliferation. The right day 6 wound stage is predicted as mostly proliferation, along with hemostasis and inflammation. All probabilities for maturation were zero. Healnet's output aligns well with the expected progression of healing stages, along with features visible in the wound bed, such as blood pooling during hemostasis and dry, diffuse edges during proliferation. Further details on the method used to process these images with Healnet are included in Supplementary Fig. S7.

Figure 5b shows the result of three multi-output linear Ridge regression models to predict the day and % re-epi. based on the fluorescent NO images. The Ridge regression algorithm is used for its effectiveness in estimating coefficients of multiple linear regression models where independent variables are correlated[47], which is expected with the variables healing day and % re-epi. In the first model (full wound —fluorescent intensity, Fig. 5b), the mean fluorescent intensity values are computed by averaging the green color channel in the wound bed, using an image manually cropped with ImageJ[48].

In the second and third regression models, we apply the $k$-means clustering algorithm to gain further insight into the spatiotemporal role of NO. Similar to the pH $k$-means clustering technique, the $x$, $y$ pixel positions and the fluorescent intensity (green color channel) are used. In this experiment, we found improved results when excluding the other (red, blue) color channels from the analysis. Like in the pH detection experiment, we use $k = 5$ clusters to segment the wound bed. Figure 5c visualizes the clustering output, where the color bars indicate the NO fluorescent intensity as a percentage of the green pixel value, which ranges from 0 to 255. The output demonstrates clusters with increasing levels of NO and highlights its non-uniform distribution throughout the wound bed. To examine the relationship between the NO and the wound stage, we use the cluster with the highest intensity fluorescent mean value to build the remaining regression models. This cluster represents the region of the wound with the highest NO expression. The second model (Highest NO cluster—fluorescent intensity, Fig. 5b) uses the mean fluorescent intensity of the selected cluster to predict the day and % re-epi. Finally, the third model (Highest NO cluster—area, Fig. 5b) uses the wound area percentage of the selected cluster for prediction. This wound area is equal to the proportionate number of pixels in the selected cluster.

To support the numerical stability of the regression, all variables are normalized between 0 and 1 as follows. The day variable is normalized

**Table 2 | Healnet wound stage probability predictions**

|  | Left wound | Right wound | Left wound | Right wound |
|---|---|---|---|---|
|  | **3-day-old** | **3-day-old** | **4-day-old** | **4-day-old** |
| Hemostasis | 0.48 | 0.98 | 1.0 | 1.0 |
| Inflammation | 0.52 | 0.02 | 0 | 0 |
| Proliferation | 0 | 0 | 0 | 0 |
| Maturation | 0 | 0 | 0 | 0 |
|  | **5-day-old** | **5-day-old** | **6-day-old** | **6-day-old** |
| Hemostasis | 0 | 0 | 0 | 0.16 |
| Inflammation | 0.01 | 0 | 0 | 0.18 |
| Proliferation | 0.99 | 1.0 | 1.0 | 0.66 |
| Maturation | 0 | 0 | 0 | 0 |

The brightfield images are used to predict the probability (0–1) of hemostasis, inflammation, proliferation, and maturation using Healnet, a pre-trained machine learning algorithm.

**Table 3 | Nitric oxide (NO) regression mean squared error (MSE)**

|  | Fluorescent intensity |  | Clustered wound —area |
|---|---|---|---|
|  | **Full wound** | **Clustered wound** |  |
| Day total MSE | 0.19 | 0.18 | 0.19 |
| Day CV MSE | 0.34 | 0.30 | 0.31 |
| % Re-epi. total MSE | 1.38% | 1.31% | 1.14% |
| % Re-epi. CV MSE | 2.73% | 2.61% | 2.00% |

MSE for NO multi-variable regression results, including the total MSE and the cross-validation (CV) MSE for each variable (day and % re-epi.). This analysis is performed for the three regression models shown in Fig. 5b.

with day 3 equal to 0.5 and day 6 equal to 1. The % re-epi. is represented as a value between 0 and 1. The fluorescent intensity (FI) values are normalized so that the minimum pixel value, 0, is set to 0 and the maximum pixel value possible, 255, is set to 1. Finally, the wound area percentage is normalized between 0 and 1. The regularization term for the Ridge regression is set to 0. We find the following coefficients for each relationship;

$$\text{day} = 0.85 * (\text{Full Wound FI}) + 0.57 \quad (1)$$

$$\% \text{ re-epi} = 0.40 * (\text{Full Wound FI}) + 0.25 \quad (2)$$

$$\text{day} = 1.08 * (\text{Clustered Wound FI}) + 0.03 \quad (3)$$

$$\% \text{ re-epi} = 0.56 * (\text{Clustered Wound FI}) - 0.04 \quad (4)$$

$$\text{day} = -1.08 * (\text{Clustered WA}) + 0.92 \quad (5)$$

$$\% \text{ re-epi} = -1.13 * (\text{Clustered WA}) + 0.51 \quad (6)$$

where FI is the average fluorescent intensity and WA is the wound area percentage of the highest intensity cluster. Figure 5b shows the relationships given the regression coefficients, which indicate a positive relationship between NO and the variables day and % re-epi for the fluorescent intensity of the full and clustered wound bed. In contrast, we observe a negative relationship between the wound area of the highest-intensity cluster and the variables day and % re-epi.

The mean squared error (MSE) for the day coefficients using all of the available data is 0.19 days and 1.4% for the % re-epi., indicating a strong predictive power by the regression. Details for the MSE calculation are provided in the "Statistics and reproducibility" section. The model's performance was further evaluated using 8-fold cross-validation to demonstrate the stability of the regression coefficients, where one sample is left out of each trained model. Across the 8 folds, the average MSE for predicting the day was 0.34 days. The average MSE for predicting the % re-epi. was 2.7%. The MSE shows a similarly strong predictive power for the clustered wound regression models and is summarized in Table 3. The output model coefficients for the Full Wound model are plotted in Supplementary Fig. S8. These results demonstrate the model's ability to accurately predict both the time post-wounding (day) and the extent of re-epithelialization based on the fluorescent NO images.

In our experimental results, shown in Fig. 5b, we observe a gradual increase in fluorescent, or NO, intensity from day 3 to 6, even with both day 5 and 6 wounds predicted to be well into proliferation. We also observe a gradual increase in NO intensity as a function of % re-epi, with the model trained on a % re-epi. range of 18.1–53.6%. This trend is reflected in the mean NO intensity of the cluster with the highest NO expression. However, we observe a gradual decrease in the wound area of the highest NO cluster as a function of both the day and the % re-epi.

Past studies suggest that NO synthesis increases rapidly during the first days of wound healing, peaks around day 3, and then gradually decreases, based on the relative numbers of the cell types that dominate NO production in wounds[42]. This is understood to align with the wound healing stages, where NO synthesis initially increases during the inflammatory phase, and decreases as the wound progresses into later stages[49]. Other studies have also measured increasing concentrations of nitrite and nitrate, stable metabolites of NO, in wound fluid until the second week post-wounding, suggesting sustained NO synthesis[50]. However, the measurements of these stable metabolites of NO do not correspond exactly to NO formation since nonenzymatic NO formation can occur[42].

Other proxy detection methods for NO have also previously included immunohistochemistry, measurement of enzyme content, peroxynitrite formation in tissue, and gene expression[42]. In contrast, DAF dyes have been proposed as a specific and sensitive method of directly measuring NO gas at the wound surface[51]. Therefore, our results may reflect a true biological phenomenon rather than experimental artifacts. Future studies could investigate the relationship between these approaches. Factors that may explain the increasing levels of NO in our study compared to past models include a limited sample size, dysregulation of NO in the wound bed, or production of NO by other cell types in the wound.

Our findings imply that as wound healing progresses, the area of the wound that highly expresses NO decreases, even though the intensity of NO within these regions remains elevated. This suggests that the spatial distribution of NO expression, rather than its overall intensity, may play a critical role in wound healing. Current literature supports the importance of NO in early wound healing, particularly during the inflammatory and proliferative phases. However, our results add a new dimension to this understanding by showing that as healing advances, NO production becomes more localized to specific areas of the wound. This spatial restriction highlights the importance of targeted NO production and could influence how NO-based therapies are developed and applied. These findings underscore the importance of not just the quantity but the localization of biochemical factors like NO in wound beds during healing[52]. Extending imaging through full wound closure would further deepen insights into the role of NO throughout wound healing. Future research could also use these images in conjunction with the histology of genetically marked endothelial cells to co-localize our findings. This analysis highlights the utility of an imaging platform that provides both brightfield and fluorescent imaging, as the wound stage can be derived from the brightfield images, while NO-specific fluorescent images give predictive power into the healing day and % re-epi. of wounds.

In this study, we presented a foundational characterization of our proposed imaging unit, including the interpretation of fluorescent dyes in vivo, leading to biological insights about the role of biomarkers in wound healing. In future iterations, the imaging unit could be adapted to support and interface with complementary systems. For instance, a multi-filter mechanism could provide a more comprehensive assessment of the wound state. Previously, pH and NO imaging have been performed sequentially to monitor both biomarkers in the same sample[53]. Given the customizability of our fabrication method, a switching mechanism could be introduced to support multiple filters within a single unit, though this introduces various engineering challenges, including size constraints for the filter exchange mechanism, durability, and mechanical stability. Finally, a bioelectronic device that interfaces with the wound could be integrated with the imaging unit. This would enable both the delivery of charged dyes and rinsing solutions to the wound bed, potentially advancing diagnostic and treatment capabilities in future wound care cases.

## Methods

### Ethics declarations: approval for animal experiments

All animal experiments were conducted under the protocol approved by the University of California Davis (UC Davis) Institutional Animal Care and Use Committee (IACUC). All methods were performed in accordance with the UC Davis IACUC guidelines and regulations. All animal studies are reported in accordance with ARRIVE guidelines. We have complied with all relevant ethical regulations for animal use.

### Hardware design and development

The imaging unit is composed of several key components: a Raspberry Pi, a custom 3D printed housing, lenses, a 21MP Arducam camera, emission filters, an excitation LED PCB, and a PDMS diffusing spacer. Supplementary Fig. S9 shows the connection diagram for power and data between the imaging unit, computing unit, and peripherals. The imaging unit has a compact square footprint measuring 3.8 cm² at the base and a height of 2.16 cm. The total object distance in the presented design is 13.9 mm. The optical configuration shown in Fig. 1b allows the resolution of features down to 24.80 μm, as demonstrated by a calibration slide in Supplementary Fig. S1.

The optical design was informed by experimental requirements for in vivo tests, including a 6mm diameter field of view, sub-millimeter resolution, and a maximum image capture time of 60 s. Lens selection was guided using an open-source optical ray tracer software to determine the optimal focal plane. Following initial simulations, an optical stage helped refine the exact object distance, as shown in Fig. 2a. The height of the 3D-printed mold used to cast the silicone diffuser can be edited to accommodate different desired object distances. This adaptability reduced the need to frequently redesign the 3D printed housing with changes to the desired object distances and has been tested successfully up to 8.8 mm. Various lens configurations have been trialed in the imaging unit. The configuration shown in Fig. 1b was used for Snarf imaging, while the configuration used for DAF-FM and Calcein AM imaging is shown in Supplementary Fig. S2.

Assembly steps for the imaging unit are illustrated in Supplementary Fig. S10. The 3D-printed housing consists of a three-part mount and a shroud. The mount secures the camera, lens, and filter components, while the shroud encloses the mount and camera, along with protecting the CMOS PCB. The three interlocking mount parts include a bottom mount, which sits atop the camera breakout board to hold the filter; a middle mount, which holds the lens and encases the top of the filter; and a top mount, which encases the lens and provides a base for the LED PCB. The mount design offers modularity and adaptability for various optical parameters, like object distance, magnification, and field of view. The designs for the mount and the camera shroud can be edited in CAD to accommodate different emission filters and object distances. The assembly is fastened to the camera board with two m2 × 16 mm screws and hex nuts. The shroud, fitted around the mount through a friction fit, features inserts to embed four threaded inserts; four m3 × 5 mm screws are inserted from the bottom of the mount at each

corner to complete the assembly. Details of these components, including specifications and part numbers, are provided in our Bill of Materials in Supplementary Table S1. The Bill of Materials for the peripherals and the computing unit are detailed in Supplementary Table S2.

**PCB LED and emission filters.** To accommodate the excitation of various fluorescent dyes, we designed a custom LED PCB to independently control three LED channels. This PCB consists of 3 driver transistors and a total of 12 LED pads. Each transistor drives four LED banks. The LEDs are powered and operated by the Raspberry Pi's DAC pins, using a pulse width modulation signal. For brightfield imaging, we use White light LEDs. LEDs with a 520 nm ± 5 nm wavelength are used to excite Snarf fluorescent dye. These LEDs are paired with a 10 × 10 mm, 645 nm bandpass emission filter (FWHM 17 nm, indicating the spectral width at half maximum transmission) to measure the fluorescent signal. LEDs with 496 nm ± 6 nm are used to excite DAF-FM and Calcein AM fluorescent dyes along with a 515 nm (FWHM 10 nm) bandpass emission filter. The filters are included in the Supplementary Table S1. The PCB components are listed in Supplementary Table S3, along with the PCB layout diagram in Supplementary Fig. S11.

**Silicone diffuser and wound proxy.** We designed and fabricated the PDMS diffuser and wound proxy using a 3D-printed mold (Supplementary Fig. S3). The diffuser serves three main purposes: it acts as a spacer to obtain the correct optical focal length, as a diffuser for the LEDs, and as an interface to protect both biological samples and the housing from contamination. PDMS was selected because it is chemically inert, biocompatible, and transparent. We incorporate a circular tunnel into the diffuser design to accommodate the protruding electronic components of the LED PCB. The PDMS body is transparent, which is critical for sample illumination as the bottom surface of the diffuser sits directly on top of the LED PCB. However, the surface of the circular tunnel is slightly textured because it is in contact with the 3D printed mold as the PDMS cures, which allows for some diffusion of the LEDs. Between uses of the camera, the PDMS diffuser is cleaned with isopropyl alcohol (IPA) and repositioned on top of the LED PCB. The PDMS diffuser adheres to the LED PCB, which is mechanically fixed to the imaging unit, as PDMS can be somewhat tacky when clean. This adhesion remains secure even when the imaging unit is inverted. After an experiment, the diffuser can be removed and replaced as needed.

To calibrate the imaging unit for pH detection, we employed a wound proxy, addressing the challenge that fluorescent signals can significantly vary with environmental changes and the condition of the imaged dye, such as dye layer height. We used PDMS, as opposed to 3D-printing the wound proxy, due to its chemical inertness and its ability to simulate the wound surface's properties, including flexibility and translucency.

A CAD model was created in Autodesk Fusion 360 to fabricate the molds for the PDMS components. The STL files were imported into the Formlabs PreForm software and printed on a Formlabs Form 3 SLA resin printer using Model V2 material. Post-printing, the molds are sonicated in IPA, rinsed off with water, dried with N2 gas, and UV-cured. They are then filled with recently mixed and degassed Sylgard PDMS (10:1 ratio) and cured in a 60 °C oven for 48 h. The PDMS pieces are demolded by running a sharp blade along the edge of the mold. The mold designs are shown in the Supplementary Fig. S3.

### Imaging procedure

Our system consists of peripherals, a computing unit or microcontroller, and the imaging unit. The peripherals (mouse, keyboard, and monitor) provide user input to the microcontroller, control the LED banks, and provide real-time image viewing from the imaging unit. The microcontroller sends image capture protocols and receives image files from the imaging unit using libcamera software. The imaging unit can be used handheld, facing up or down, within a 3 ft cable length from the peripherals and within a 15 cm cable length from the computing unit.

**Snarf detection**. The LED PCB features 520nm excitation LEDs, and the imaging unit houses a 580 nm bandpass filter to perform the pH imaging experiments. For pH detection, we use the fluorescent dye 5-(and-6)-Carboxy SNARF™-1 (Snarf) from Thermofisher. The Snarf dye is diluted in pH buffer to validate and calibrate the imaging unit, and in DI water to estimate the in vivo wound bed pH. The wound proxy was created to capture a consistent fluorescent signal using a casting process with PDMS (Supplementary Fig. S3). The body of the PDMS wound proxy is a rectangular prism that matches the dimensions of a standard microscope slide. The wound is represented by a circular indentation in the PDMS with a 6 mm diameter and 0.6 mm depth. The PDMS wound proxy is placed on top of a glass microscope slide during imaging, shown in Fig. 3a.

Tris buffer is used to obtain an exact pH by adding HCl and NaOH as needed; 25 μL of Snarf dye is diluted in 975 μL of Tris buffer of pH 5.9, pH 7.1, pH 8.0, and pH 9.0 to make a 50 μM concentration of Snarf. The pH of the Tris buffer was measured using a benchtop pH meter (Thermo Scientific Orion™ Versa Star Pro™) and probe (Thermo Scientific Orion™ ROSS Ultra™ Low Maintenance pH/ATC Triode™). Twenty microliters of the dilution for each pH buffer was pipette into the proxy wound. This sample was placed into the commercial fluorescent microscope (Keyence BZ-X710) and imaged with a 590 and 640 nm filter cube (Chroma SNARF CUBE 1 590/33 and CUBE 2 640/30). Each pH level was also imaged with our unit to capture the calibration images, as shown in Fig. 3c.

Typically, Snarf is used as a ratiometric dye to account for factors such as dye concentration, variations in illumination intensity, and photobleaching. To use a single emission wavelength with our imaging unit, we pipette the dye to ensure a precise concentration, turn off the lights to have a dark room, and use the PCB LED for consistent illumination across samples.

For in vivo pH detection, we prepared a solution of 25 μL of Snarf diluted in 975 μL of DI Water. On day 0 of the experiment, 6 mm diameter circular wounds were generated and a circular ring was sutured on. The animals were sacrificed on day 3. Each day, the mice were first anesthetized, weighed, and imaged using a commercial cell phone camera. While anesthetized, 20 μL of the dilution was pipette onto the wound bed. Next, we aligned the PDMS diffuser/spacer of the imaging unit on top of the sutured ring and captured a z-stack of 11 images which were saved to the microcontroller. After the experiment, images were transferred and analyzed using OpenCV and Python3 in a Jupyter Notebook. Further experimental details are available in the IACUC protocol.

**DAF-FM and Calcein AM detection**. To image the DAF-FM (DAF-FM Diacetate (4-amino-5-methylamino-2′,7′-difluorofluorescein diacetate), Thermofisher) and Calcein AM (Thermofisher) fluorescent dyes, the LED PCB housed 496 nm excitation LEDs, and the imaging unit a 515 nm bandpass filter. For the in vivo NO detection experiment, 10 μL of 5 μM concentration dye was applied to the wound bed with a pipette. After a 5-min waiting period, excess moisture was removed with gauze, and single images were captured with the imaging unit. These images were again transferred for post-processing and analysis. In this experiment, different mouse wounds were imaged on sequential days since the wound was generated. For instance, the Left Wound images shown in Fig. 5a are from a wound generated 3 days prior, while the Right Wound images are from a different animal, with the wound generated 4 days prior.

## Statistics and reproducibility

The $R^2$ values for the regression results were computed by squaring the Pearson correlation coefficient (r-value) obtained from the linregress function in the scipy.stats module (Python)[35]. The chi-squared test for variance to determine the one-sided p-value was computed as $\chi^2 = (n-1) * s^2/\sigma^2$, where $n$ is the sample size, $s^2$ is the sample variance, and $\sigma^2$ is the target or comparison variance. The MSE reported in the analysis of the NO results was calculated using the mean_squared_error function from scikit-learn[54]. The function compares the predicted values ($\hat{Y}_i$) from the linear regression to the training data ($Y_i$), with the number of samples, $n = 8$, as $\mathrm{MSE} = \frac{1}{n}\sum_{i=0}^{n}\left(Y_i - \hat{Y}_i\right)^2$.

The experiments used a minimal number of samples (two wounds per day) to serve as a proof-of-concept, aiming to generate initial data and test the feasibility of the imaging techniques presented. In the pH study, three wounds were imaged over day 0–3 post-wounding. In the NO study, two wounds per day were imaged on days 3–6 post-wounding. As this was a preliminary investigation, replication was not attempted.

## Animal experiments

Healthy, wildtype C57BL/6J mice from Jackson lab, Sacramento, CA (males, 40 weeks old, 34–49 g) were used. They were acclimated and supplied with DietGel 93M (ClearH2O) and soaked chow to maintain their body weight for 1 week before the experiment started. The mice were weighed and shaved 1–3 days prior to the surgery. Wounding groups were randomly assigned by weight. On the day of surgery (day 0), the animals were anesthetized with 1–5% isoflurane inhalation, saline, and analgesics were injected, and the back skin was prepared with betadine and alcohol washes. One wound was created on each side of the spine by suturing silicon splint rings (16 mm outer and 10 mm inner diameters) on the intended location. A 6 mm biopsy punch generated a full-thickness, excisional wound with the silicon splint to control wound contraction on each mouse. The wounds were covered with a vapor-permeable secondary dressing (Tegaderm). Examination of the wounds and dressings are performed daily, starting on the surgery day (day 0). On the final experiment day, the animals were euthanized through cervical dislocation with deep anesthesia. Our protocol specified exclusion and euthanasia for any animal that lost more than 20% weight, but none did. Housing in a temperature and daylight-controlled environment with food and water ad libitum, along with daily gross examination and 24/7 vet care, was provided. Post-operative analgesics were given every 12 h for 48 h, and signs of pain were monitored during the experiment. Confounding factors were not controlled due to the proof-of-concept experimental design. The wound surgery and fluorescent imaging were not blinded due to the nature of the technology, necessitating the operating investigator to be aware of the experimental protocol. The histology for the re-epithelialization rates was a blinded measure.

## Histology

At the end of each experiment, after euthanizing the mice, the circular wounds were bisected at their widest part to preserve the relative re-epithelialization to wound-width ratio. Half-wound pieces were fixed for 24 h in a buffered solution of 4% paraformaldehyde and moved to 70% ethanol for 24 h for dehydration. The tissues were processed for FFPE histology and embedded into paraffin blocks (Tissue Tek processing and embedding stations, Sakura Finetek, Torrance, California), cut into 5 μm thick sections using a microtome, and the sections were placed onto glass slides. Once dried, the sections were stained using a standard H&E protocol to assess re-epithelialization. Stained sections were imaged using a BZ-9000 inverted microscope, and the images were scored in the BZ-II Analyzer software (Keyence, Osaka, Japan). The wound edges were identified using the hair follicles innermost towards the wound center. Epithelial ingrowth was measured along the basal keratinocyte layer from the innermost follicle to the tip of the epithelial tongue, on each side of the wound. The total width of the wound was measured along the surface of the granulation tissue between the two innermost follicles, and % re-epi. was calculated as the percentage of the total width that was covered by the epithelial ingrowth[55]. Tissue samples are shown in the Supplementary Fig. S12.

## Reporting summary

Further information on research design is available in the Nature Portfolio Reporting Summary linked to this article.

## Data availability

The data generated and analyzed during this study are available from the corresponding author upon reasonable request. The numerical source data for the plots are available in Supplementary Data 1.

## Code availability

Custom code used for the pH and NO analysis is available from the corresponding author upon reasonable request for research purposes. The Healnet wound stage prediction model is open source and available on GitHub and on Zenodo[56].

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

## Acknowledgements
This project is supported by the Defense Advanced Research Projects Agency (DARPA) through Cooperative Agreement Number D20AC00003 awarded by the U.S. Department of the Interior (DOI), Interior Business Center.

## Author contributions
The manuscript was prepared by M.Tebyani and revised by M.R., R.R.I., and M.Teodorescu. The camera design was developed by G.K., M.Tebyani, D.H., A.Sher, and M.Teodorescu. The system was fabricated by G.K., M.Tebyani, W.S.H., P.B., C.H., and A.B. Experiments were supported by M.Tebyani, G.K., W.S.H., H.D., T.N., H.H., and H.Y. Results were analyzed by M.Tebyani, G.K., H.D., H.C., N.N., A.G., and A.Spaeth. E.A. coordinated experiments. M.R., M.Teodorescu, M.Z., and R.R.I. secured the funding and directed the research. All authors reviewed the manuscript.

## Competing interests
The authors declare no competing interests.
