## [Transparent Peer Review file · Communications Biology]

A modular fluorescent camera unit for wound imaging

Corresponding Author: Dr Maryam Tebyani

Version 0:

Reviewer comments:

Reviewer #1

(Remarks to the Author)

The authors designed a novel, modular, hand-held fluorescent microscope featuring rapid set-up and sub-millimeter resolution for real-time biological analysis. This study giving a good example of combining cross-disciplinary strategies and techniques in both structures and functions to work towards improved results. The article is suitable for publication after the following concerns are addressed:

1. Figure 1e shows bright-field and DAF-FM fluorescent images from mouse back wounds. Could auxiliary lines be used to demarcate the wound area, and could a digital camera be added to display the wound size?
2. The pixel resolution of Figure 4 appears to be different from the other images, resulting in a blurry and unclear graphic. Could it be replaced with a high-resolution image?
3. Regarding the pH mapping in Figure 4c of the wound, does the graphic represent the entire wound size? Why does the same wound appear smaller on Day 0 imaging and then larger on Day 1? Typically, wounds should gradually decrease in size over time; why does it appear to enlarge in the graphic?
4. It is recommended to monitor the pH and NO levels of the wound until the wound is completely healed.

Reviewer #2

(Remarks to the Author)

Summary:

The authors have developed a unique handheld modular wound imaging unit that can be utilized for in situ imaging of biological tissue to analyze real-time the wound healing response. The authors were able to correlate the imaging units ability to detect changes in pH and NO levels, which are important regulators of wound healing progression, at different time points to known phases of wound healing, such as epithelialization. These studies were performed in vitro and validated in vivo with a mouse model where the authors were able to uniquely identify changes in wound pH levels and a temporal-spatial relationship of NO levels during wound healing. The methodology of the study is well described, although the statistics are less clear. Overall the imaging unit appears to be user friendly, although it would be beneficial to be able to have more than one wavelength capability within the unit. This system could prove to be an effective approach for evaluating wound healing within research to further support and identify temporal-spatial aspects of the wound healing phases in situ. However, it is not clear clinically if this would be utilized with patients if it requires perpetual addition of fluorescent labeling and imaging. By addressing the comments/questions below, this reviewer would support the publication of this work.

Comments:

In figure 2, is it possible for the authors to use different fluorophores for the Calcein-AM and DAF-AM in order to allow for an overlay of viable cells with the NO⁺ cells to ensure that the NO⁺ cells are in fact viable cell populations given the propensity of DAF-FMN to stain dead or dying cell populations (e.g., interchange the excitation/emission filter while in place given the ability of only being able to use one wavelength at a time).

In figure 3c, is it possible for the authors to include fluorescent data that includes a pH range of 4, given the role of this pH range being important for certain phases of wound healing (i.e., epithelialization)? Would pH values outside of 5-9 be

outside the linear range for this methodology? Do the authors feel like not including pH values outside this range is missing any critical information?

The emission peak of carboxy-SNARF at pH 6.0 is at wavelength ~580nm, while that of pH 9.0 is at ~640nm and therefore a ratio of intensity is typically taken between the two wavelengths. In figure 3b and 3c, the authors use 640nm as the emission wavelength for calculating pH based on FI. Given this handheld imaging unit only utilizes one wavelength (i.e., 640nm), do the authors feel there is discrepancies from the single wavelength data and the true pH values? Will future iterations of this device be able to allow for multiple wavelength measurements?

What are the authors thoughts on the 2-3 fold increase in error for their imaging unit when calculating pH at higher pHs (~8) compared to lower pHs (~5-7) when the emission wavelength of 640nm used by the imaging unit should ideally be more accurate at higher pH values given the pH-dependent wavelength shift in peak emission for carboxy-SNARF?

For figure 4b, what are the authors thoughts on the initial positive deflection (Day 1) in the "Clusters" for wounds 1 and 2, but negative deflection for wound 3? There appears to be significant variability in the clustering and pH values between the different wound replicates. How would the authors explain this variability? Were the authors able to perform any statistics to explain to differences seen?

In figure 5a (and 1e), can the authors provide a non-stained fluorescent control (without DAF-FM) for comparison (could provide within the supplemental data)?

In figure 5c the figure legend/text states the color bars indicate pH but the scale of the number is -80 to -200, is this accurate to say that the color indicates pH? If so, is there a way for the authors to convert this data to match the true pH rather than an arbitrary unit range?

For the H&E cross-section images measuring epithelialization of the wounds, given the circular nature of the wounds, the relative tongue-distance/wound-width ratio may change if not taken across the widest (centerpoint) of the wounds consistently. Were the authors able to account for this when they were measuring epithelialization (i.e., by taking from the same relative regions within each set of wounds at the largest diameter)?

As the authors state, NO is often thought to peak around Day 3 post-wounding (depending on the context of the wounds), but within this study there is a continuous rise in fluorescent intensity (i.e., NO levels) through Day 6. The authors state this is likely due to discrepancies related to artifact of traditionally utilized methods of measuring NO and DAF-FM being more sensitive. This may be valid but it would strengthen the authors claims if they were able to provide data from the "less sensitive" modalities demonstrating a Day 3 peak as traditionally seen in literature (as they discuss in the text).

Reviewer #3

(Remarks to the Author)

This work reports the development of hand-held and modular imaging units consisting of five fundamental components for biological imaging. The imaging unit is capable of performing various imaging tasks, including brightfield and fluorescent imaging. The authors demonstrated its use for pH and NO detection, both in vitro and in vivo, along with an analysis of the detection using machine learning. While the manuscript presents an interesting approach and application, several major concerns are outlined in the comments below. I recommend publishing this work only after all comments are thoroughly and adequately addressed:

1)What is the rationale for using the THP-1 cell line? Why not use skin or wound-related cell lines that are more relevant to the scope of this work , such as fibroblasts, endothelial cells, and others?

2)On page 5, pH values should be compared and validated using a gold-standard device, such as pH meter or any other method used today for pH monitoring in wounds, in addition to the commercial microscope used in the experiments to ensure the accuracy of pH measurements.

3)There are several typos and errors throughout the manuscript that need to be corrected. For example, in the caption of Table 3 on page 11, "Figure ??b" should be fixed.

4)A major missing aspect is imaging for all biomarkers simultaneously. The authors showed separate imaging and characterization for pH and NO, but for the application to be more applicable and novel, a single imaging approach providing multiple channels at once and delivering comprehensive wound status information in one image would be more effective. The authors should include this experiment to demonstrate the integration of all presented values. If this is not feasible with the current application, the authors should explain the limitations of their technology. As the manuscript stands and ends, it feels incomplete. To wrap up the discussion, the authors should address this issue and consider mentioning it as part of current or future work.

Version 1:

Reviewer comments:

Reviewer #1

(Remarks to the Author)

The authors designed a novel, modular, hand-held fluorescent microscope featuring rapid set-up and sub-millimeter

resolution for real-time biological analysis. This study giving a good example of combining cross-disciplinary strategies and techniques in both structures and functions to work towards improved results.

Reviewer #2

(Remarks to the Author)

The authors have appropriately taken the time to provide the requested revisions and/or provide an appropriate response/clarification as to why they were unable to. The authors made amendments to the manuscript text where asked to and responses were sufficient in detail and now provide the necessary content. At this point, this reviewer has no further concerns with the manuscript and feels it is acceptable for publication.

Reviewer #3

(Remarks to the Author)

The authors have adequately addressed all my comments. I now recommend the paper for publication.

Please find our response to the comments from reviewers below. Figure 1 and Figure 5 have been modified, and the main body and supplementary drafts returned include blue-colored text to specify text edits.

Reviewer #1

The authors designed a novel, modular, hand-held fluorescent microscope featuring rapid set-up and sub-millimeter resolution for real-time biological analysis.

This study giving a good example of combining cross-disciplinary strategies and techniques in both structures and functions to work towards improved results. The article is suitable for publication after the following concerns are addressed:

1. Figure 1e shows bright-field and DAF-FM fluorescent images from mouse back wounds. Could auxiliary lines be used to demarcate the wound area, and could a digital camera be added to display the wound size?

*Response: Thank you for your comments regarding Figure 1e. We would like to clarify that the brightfield and DAF-FM fluorescent images shown in Figure 1e are magnified views (100 μ m scale) that display only tissue within the wound area. The complete brightfield and fluorescent wound images are shown respectively in Figure 5a_{ii} (left) and Figure 5b_{iii} (right). We have edited **Figure 1** to feature an inset showing where the magnified view is cropped from Figure 1a, and have edited the **Figure 1 caption and Imaging (lines 81-82)** section of the main body to further clarify this.*

2. The pixel resolution of Figure 4 appears to be different from the other images, resulting in a blurry and unclear graphic. Could it be replaced with a high-resolution image?

*Thank you for this observation regarding the resolution of the images in Figure 4. We believe the lower resolution is not due to image quality issues but rather reflects the inherent nature of pH imaging. We do not expect to see specific structures in these images because H⁺ ions diffuse to adjacent tissues, so pH imaging would not be localized to specific structures. We have clarified this point in the **Imaging - pH Detection (lines 177-180)** section of the main body to help with the interpretation of these results.*

3. Regarding the pH mapping in Figure 4c of the wound, does the graphic represent the entire wound size? Why does the same wound appear smaller on Day 0 imaging

and then larger on Day 1? Typically, wounds should gradually decrease in size over time; why does it appear to enlarge in the graphic?

*Thank you for this comment. The graphics do represent the entire wound size. While wounds generally decrease in size over time, previous literature has reported increases in the first 3 to 4 days of healing in similar wound models. We have added text in the **Imaging - pH detection (lines 200-201)** section of the main body to address this comment, including a reference to a journal on automatic wound detection and size estimation, which demonstrates an increase in wound size between Day 0 and Day 1 [1].*

[1] Carrión H, Jafari M, Bagood MD, Yang Hy, Isseroff RR, et al. (2022) Automatic wound detection and size estimation using deep learning algorithms. PLOS Computational Biology 18(3): e1009852. doi:10.1371/journal.pcbi.1009852

4. It is recommended to monitor the pH and NO levels of the wound until the wound is completely healed.

*We agree that monitoring pH and NO levels until complete healing would provide valuable insights. Unfortunately, our study was designed with a defined experimental endpoint that considered data collection with other constraints, such as limited resources. As expanding the monitoring period to capture the complete healing process is recommended, we have included longitudinal studies as an area for future investigation in both the **Imaging - pH Detection (lines 233-235)** and **NO (lines 380-382)** sections of the main body.*

Reviewer #2

Summary:

The authors have developed a unique handheld modular wound imaging unit that can be utilized for in situ imaging of biological tissue to analyze real-time the wound healing response. The authors were able to correlate the imaging units ability to detect changes in pH and NO levels, which are important regulators of wound healing progression, at different time points to known phases of wound healing, such as epithelialization. These studies were performed in vitro and validated in vivo with a mouse model where the authors were able to uniquely identify changes in wound pH levels and a temporal-spatial relationship of NO levels during wound healing. The methodology of the study is well described, although the statistics are less clear. Overall the imaging unit appears to be user friendly, although it would be beneficial to be able to have more than one wavelength capability within the unit. This system could prove to be an effective approach for evaluating wound healing within

research to further support and identify temporal-spatial aspects of the wound healing phases in situ. However, it is not clear clinically if this would be utilized with patients if it requires perpetual addition of fluorescent labeling and imaging. By addressing the comments/questions below, this reviewer would support the publication of this work.

1. In figure 2, is it possible for the authors to use different fluorophores for the Calcein-AM and DAF-AM in order to allow for an overlay of viable cells with the NO⁺ cells to ensure that the NO⁺ cells are in fact viable cell populations given the propensity of DAF-FMN to stain dead or dying cell populations (e.g., interchange the excitation/emission filter while in place given the ability of only being able to use one wavelength at a time).

*Thank you for this comment. We acknowledge the limitation that DAF-FM can enter and stain dead cells. According to the DAF-FM data sheet from Thermofisher [2], this dye has a fluorescence quantum yield of ~0.005, which increases to ~0.81, about 160-fold, after reacting with NO. Therefore, we believe that the fluorescence emission we have imaged primarily reflects NO concentration, the focus of our study, rather than non-specific staining. We have added text to the **Imaging - Nitric Oxide (NO) and Wound Stage Detection (lines 257-260)** section of the main body to describe this limitation and to describe that while DAF-FM does fluoresce once it enters a dead cell, we expect the fluorescent signal to be dominated by reaction with nitric oxide due to the significant difference in quantum yields.*

[2] Thermo Fisher Scientific. (2018). Nitric Oxide Indicators: DAF-FM and DAF-FM Diacetate (User Guide). Doc. Part No. MP23841, Pub. No. MAN0002276 Rev. A.0.

2. In figure 3c, is it possible for the authors to include fluorescent data that includes a pH range of 4, given the role of this pH range being important for certain phases of wound healing (i.e., epithelialization)? Would pH values outside of 5-9 be outside the linear range for this methodology? Do the authors feel like not including pH values outside this range is missing any critical information?

Thank you very much for your comment regarding the valid range of the pH dye. We acknowledge that pH 4 may occur as wounds return to the acidic pH of healthy skin through epithelialization, and that lowering the acidity of a wound to pH 4 or below may be a valuable therapeutic treatment. However, we do not expect a pH below 5 for the

*focus of our study, which monitors the wound pH between Day 0 to Day 3 of healing. We performed a calibration down to pH 4 and concluded that it was outside the range of the dye's response that could be linearly characterized. We have included the pH 4 calibration result for the 640nm wavelength captured by both the commercial microscope and our imaging unit in **Supplementary Figure 7a** and updated the **Imaging - pH Detection (lines 141-143)** section in the main body accordingly.*

3. The emission peak of carboxy-SNARF at pH 6.0 is at wavelength ~580nm, while that of pH 9.0 is at ~640nm and therefore a ratio of intensity is typically taken between the two wavelengths. In figure 3b and 3c, the authors use 640nm as the emission wavelength for calculating pH based on FI. Given this handheld imaging unit only utilizes one wavelength (i.e., 640nm), do the authors feel there is discrepancies from the single wavelength data and the true pH values? Will future iterations of this device be able to allow for multiple wavelength measurements?

*We acknowledge potential discrepancies with single-wavelength compared to the ratiometric measurements, for which we have reported error rates in Table 1. Although adding a two-filter system introduces engineering challenges for our hand-held portable device, we believe it would be feasible with our approach in future work. We have included this limitation and future work for a two-filter system in the **Imaging - pH detection (lines 164, 235-237)** section and in the **Imaging - Nitric Oxide (NO) and Wound Stage Detection (lines 393-396)** section of the main body.*

4. What are the authors thoughts on the 2-3 fold increase in error for their imaging unit when calculating pH at higher pHs (~8) compared to lower pHs (~5-7) when the emission wavelength of 640nm used by the imaging unit should ideally be more accurate at higher pH values given the pH-dependent wavelength shift in peak emission for carboxy-SNARF?

Thank you for pointing out this discrepancy in the error rates of the pH calibration. We believe the increased error at higher pH values (around 8) compared to lower pH values (5-7) is primarily due to our linear approximation method rather than the quality of the underlying data.

As shown in Figure 3b and confirmed by the carboxy-SNARF data sheet, the change in fluorescent emission between pH 7-8 is greater than between pH 8-9. This means that despite using a 640nm emission wavelength, the signal change per pH unit decreases at higher pH values, resulting in reduced measurement precision in that range. In the future,

*this discrepancy could be addressed by using additional pH values during calibration or with a two-filter system, which we have also detailed in the section. We appreciate this analysis, which has helped us better analyze our system's performance, and have added text to the **Imaging - pH detection (lines 157-163)** section in the main body to clarify this.*

5. For figure 4b, what are the authors thoughts on the initial positive deflection (Day 1) in the “Clusters” for wounds 1 and 2, but negative deflection for wound 3? There appears to be significant variability in the clustering and pH values between the different wound replicates. How would the authors explain this variability? Were the authors able to perform any statistics to explain to differences seen?

Thank you for this observation. The variability in the pH value deflection directions and clustering likely reflects biological factors unique to each wound from the in vivo model. While absolute pH values are understood to vary between healing (and non-healing) wounds [3], differences in deflection direction suggest variations in the timing of inflammatory processes or other biological factors. The bar plots showing elements per cluster demonstrate corresponding differences in wound structure and composition (as represented by cluster distributions). Clustering variability may be due to differences in the wound shape between samples and days.

We performed a chi-squared test to determine the statistical significance of the center cluster having a higher variance than the other clusters, which resulted in a one-sided p-value of 0.031.

*We have added additional text to the **Imaging - pH Detection (lines 193-199)** section in the main body to acknowledge the variability in the clustering results (deflection direction and elements per cluster). We have also added to this section (**lines 211-215**) the statistical test performed to analyze the spread of the magnitude change in pH.*

[3] Gethin, G. (2007). The significance of surface pH in chronic wounds. Wounds UK 3.

6. In figure 5a (and 1e), can the authors provide a non-stained fluorescent control (without DAF-FM) for comparison (could provide within the supplemental data)?

We agree that a non-stained fluorescent control in Figure 5a (and 1e) would be valuable. Unfortunately, due to IACUC considerations for our in vivo model, we did not include non-stained controls in the NO experiment. The time window for imaging was brief, as

*there is a time limit for which the animals can be anesthetized. We could only use the imaging unit twice per wound during in vivo experiments. The in vivo DAF-FM images were captured after a brightfield image with our unit. We have included this limitation in the **Imaging - Nitric Oxide (NO) and Wound Stage Detection (lines 266-267)** section of the main body.*

7. In figure 5c the figure legend/text states the color bars indicate pH but the scale of the number is -80 to -200, is this accurate to say that the color indicates pH? If so, is there a way for the authors to convert this data to match the true pH rather than an arbitrary unit range?

*We thank the reviewer for this comment, as Figure 5c shows the NO concentration, not the pH. This was a typo in the caption, which has been corrected. We have changed the range to a percentage to signify that the color bars represent a relative range of NO fluorescent intensity in **Figure 5c**, along with updating the **Figure 5c caption** and the **Imaging - Nitric Oxide (NO) and Wound Stage Detection (lines 313-314)** section in the main body.*

8. For the H&E cross-section images measuring epithelialization of the wounds, given the circular nature of the wounds, the relative tongue-distance/wound-width ratio may change if not taken across the widest (centerpoint) of the wounds consistently. Were the authors able to account for this when they were measuring epithelialization (i.e., by taking from the same relative regions within each set of wounds at the largest diameter)?

*Thank you for this observation. Yes, we bisected the circular wound from the widest part of the wound for histology, so the measurement was the widest part for re-epithelialization. A description was added in the **Methods - Histology (lines 543-544)** section to clarify this.*

9. As the authors state, NO is often thought to peak around Day 3 post-wounding (depending on the context of the wounds), but within this study there is a continuous rise in fluorescent intensity (i.e., NO levels) through Day 6. The authors state this is likely due to discrepancies related to artifact of traditionally utilized methods of measuring NO and DAF-FM being more sensitive. This may be valid but it would strengthen the authors claims if they were able to provide data from the “less

sensitive” modalities demonstrating a Day 3 peak as traditionally seen in literature (as they discuss in the text).

Thank you for your comment. Past studies suggest that NO synthesis, rather than NO level, increases rapidly during the first days of wound healing, peaks around Day 3, and then gradually decreases, based on the relative numbers of the cell types that dominate NO production in wounds [4]. Other studies have measured increasing concentrations of nitrite and nitrate, stable metabolites of NO, in wound fluid until the second week post-wounding, suggesting sustained NO synthesis [5]. However, the measurements of these stable metabolites of NO do not correspond exactly to NO formation since nonenzymatic NO formation can occur [4], and are not directly comparable to our method of imaging NO gas on the wound surface. Given the methodological differences, our results may provide complementary information to existing literature rather than contradicting it, and future studies could investigate the relationship between these approaches.

*We have clarified these points in the **Imaging - Nitric Oxide (NO) and Wound Stage Detection (lines 354-370)** section of the main body.*

*[4] Witte, MB, Barbul, A. (2002). Role of nitric oxide in wound repair. *The American Journal of Surgery*, 183(4), 406–412. doi:10.1016/s0002-9610(02)00815-2*

*[5] Schäffer, M. R., Tantry, U., Van Wesep, R. A. & Barbul, A. (1997). Nitric Oxide Metabolism in Wounds. *Journal of Surgical Research* 71, 25–31.*

doi:10.1006/jsre.1997.5137

Reviewer #3:

This work reports the development of hand-held and modular imaging units consisting of five fundamental components for biological imaging. The imaging unit is capable of performing various imaging tasks, including brightfield and fluorescent imaging. The authors demonstrated its use for pH and NO detection, both in vitro and in vivo, along with an analysis of the detection using machine learning. While the manuscript presents an interesting approach and application, several major concerns are outlined in the comments below. I recommend publishing this work only after all comments are thoroughly and adequately addressed:

1. What is the rationale for using the THP-1 cell line? Why not use skin or wound-related cell lines that are more relevant to the scope of this work, such as fibroblasts, endothelial cells, and others?

*Thank you for your comment regarding a relevant cell line in our calibration of NO detection. Our rationale for using the THP-1 cell line was that they have been used to study NO along with monocyte/macrophage function [6], which corresponds to the inflammatory response we expected to observe in the wounds on Days 3-6 of healing during the NO detection in vivo experiment. In the cell imaging experiments, our focus was on modeling our system's performance in measuring NO as opposed to modeling a wound in vitro. We have added text in the main body to describe our reasoning behind the THP-1 cell line in the **Imaging - Cell Imaging (lines 105-107)** section of the main body, including the additional reference.*

*[6] Chanput W, Mes JJ, Wichers HJ. (2014). THP-1 cell line: an in vitro cell model for immune modulation approach. *Int Immunopharmacol.* 23(1):37-45. doi: 10.1016/j.intimp.2014.08.002. Epub 2014 Aug 14. PMID: 25130606.*

2. On page 5, pH values should be compared and validated using a gold-standard device, such as pH meter or any other method used today for pH monitoring in wounds, in addition to the commercial microscope used in the experiments to ensure the accuracy of pH measurements.

Thank you for commenting on the need for a gold-standard validation for measuring the pH in the wounds. pH values in acute wounds have been studied extensively; a summary of the currently available data for typical acute wounds ranges from high pH 5 to mid-pH 6 through early healing [7], and a mean pH value of 7.44 throughout healing [8]. These reports fall well within the range of pH values that we have found in our results.

*We have added text to the **Imaging - pH Detection (lines 190-193)** section of the main body to reflect that the pH values recorded with our device are well aligned with ranges commonly reported and accepted in the literature. Comparing the imaged wound pH to an average value obtained using a traditional measurement method to understand the relationship between these approaches has been included as a point for future work, also in the **Imaging - pH Detection (lines 231-233)** section.*

[7] Schneider, L.A., Korber, A., Grabbe, S. et al. (2007). Influence of pH on wound-healing: a new perspective for wound-therapy?. *Arch Dermatol Res* 298, 413–420. doi:10.1007/s00403-006-0713-x

[8] Sim P, Strudwick XL, Song Y, Cowin AJ, Garg S. (2022). Influence of Acidic pH on Wound Healing In Vivo: A Novel Perspective for Wound Treatment. *Int J Mol Sci.* 23(21):13655. doi: 10.3390/ijms232113655. PMID: 36362441; PMCID: PMC9658872.

3. There are several typos and errors throughout the manuscript that need to be corrected. For example, in the caption of Table 3 on page 11, "Figure ??b" should be fixed.

We have fixed the broken link and have proofread the manuscript to address typos and errors throughout.

4. A major missing aspect is imaging for all biomarkers simultaneously. The authors showed separate imaging and characterization for pH and NO, but for the application to be more applicable and novel, a single imaging approach providing multiple channels at once and delivering comprehensive wound status information in one image would be more effective. The authors should include this experiment to demonstrate the integration of all presented values. If this is not feasible with the current application, the authors should explain the limitations of their technology. As the manuscript stands and ends, it feels incomplete. To wrap up the discussion, the authors should address this issue and consider mentioning it as part of current or future work.

*Thank you for your comment regarding the need for simultaneous biomarker monitoring using a multi-filter system. We agree that this would strengthen the impact of this work, but is not feasible in the presented study, which focuses on characterizing and understanding the spatiotemporal distribution of individual biomarkers. While a multi-filter system would introduce significant engineering challenges appropriate for future research, we believe the fabrication approach we have proposed could be used to construct a multi-filter version of the imaging unit. We have renamed the Results section to **Results and discussion**, and added a discussion to address the importance and need of this future work in the **Imaging - Nitric Oxide (NO) and Wound Stage Detection (lines 389-396)**, and in the **Imaging - pH Detection (lines 235-238)** section of the main body.*